# Syntactic Awareness and Reading Comprehension in Emergent Bilingual Children

**Diana Burchell** [1,*] , **Kathleen Hipfner-Boucher** [1] , **S. Hélène Deacon** [2] , **Poh Wee Koh** [1] and **Xi Chen** [1]

1  Ontario Institute for Studies in Education, University of Toronto, Toronto, ON M5S 1A1, Canada
2  Faculty of Science, Dalhousie University, Halifax, NS B3H 4R2, Canada
*  Correspondence: diana.burchell@mail.utoronto.ca

**Abstract:** The present study investigated the role of syntactic awareness in reading comprehension among English–French bilinguals learning French as an additional language in Canadian French immersion programs. We examined the direct effect of French syntactic awareness on French reading comprehension as well as the indirect effects mediated through French word reading and French vocabulary. We further examined the extent to which English syntactic awareness contributed to French reading comprehension through cross-language transfer, again considering both the direct effect and the indirect effects through French word reading and French vocabulary. Mediation analyses indicated that, within French, the relationship between French syntactic awareness and French reading comprehension was fully mediated by both French word reading and French vocabulary. In contrast, English syntactic awareness contributed directly to French reading comprehension. Finally, French word reading partially mediated the relationship between English syntactic awareness and French reading comprehension. Our study suggests that children who learn French as an additional language rely on word reading and vocabulary, in addition to French syntactic awareness, to comprehend French texts. Given that English is French immersion children's stronger language, they use English syntactic awareness to support French reading comprehension both directly and indirectly through French word reading.

**Keywords:** syntactic awareness; English–French bilingualism; reading comprehension





## 1. Introduction

Reading comprehension is a complex process that relies upon many underlying abilities. One of these skills is syntactic awareness (SA), defined as the ability to critically analyze and manipulate the order of words in a sentence (Durgunoğlu 2002; Gombert 1992; Mackay et al. 2021). Current theories such as the Reading Systems Framework (RSF; Perfetti and Stafura 2014) position syntactic awareness as a direct support to reading comprehension through the mechanism of parsing. Studies have shown that syntactic awareness impacts reading comprehension among monolingual (Bowey 1986; Brimo et al. 2018; Deacon and Kieffer 2018; Muter et al. 2004; Browne Rego and Bryant 1993) and bilingual children (Sohail et al. 2022; Lefrançois and Armand 2003; Siu et al. 2016; Tong and McBride 2017; Yeung et al. 2011, 2013). However, to the best of our knowledge, only Sohail et al. (2022) has examined this relationship among English–French bilingual children, and the study involved emerging bilinguals in the first grade. In that study, as in our own, participants were drawn from publicly funded French immersion (FI) programs in Canada. FI is predicated on additive bilingualism, namely, the development of proficiency in oral and written French as an additional language while developing proficiency in English, the societal language. Currently, approximately 500,000 children are enrolled in FI programs nationwide (Canadian Parents for French 2022). To optimize reading outcomes among these children, it is important that the mechanisms underlying their learning be identified so as to inform evidence-based policy and teaching practices.

In addition to a direct effect on reading comprehension, a small number of studies suggest that syntactic awareness may exert an indirect effect on reading comprehension through word reading, as children use syntactic cues to aid the partial decoding of challenging words (Sohail et al. 2022; Bowey 1986; Tong et al. 2021; Tunmer et al. 1989). At the same time, syntactic awareness has been theorized to mediate vocabulary learning through the mechanism of syntactic bootstrapping (e.g., Babineau et al. 2020). Syntactic bootstrapping allows learners to use syntactic context to assign a novel word to its grammatical category, enabling them to infer the word's meaning (Babineau et al. 2020). This mechanism may explain the indirect relationship between syntactic awareness and reading comprehension through vocabulary observed in a study involving adults (Guo et al. 2011). However, no previous studies have investigated a possible indirect relationship between syntactic awareness and reading comprehension through vocabulary in children. Thus, the first two objectives of our study are to examine the direct effect of French syntactic awareness on French reading comprehension as well as indirect effects mediated through French word reading and French vocabulary in English–French bilingual children.

Extensive research has demonstrated that, for bilingual children, metalinguistic skills developed in one language enhance reading in the other through cross-language transfer (e.g., Chung et al. 2019; Koda 2007; Koda and Zehler 2008). A small number of studies have provided evidence supporting the crossover effects of syntactic awareness on reading comprehension (Sohail et al. 2022; Siu and Ho 2015). However, the small body of research was conducted on diverse bilingual populations (e.g., Chinese–English bilinguals, Spanish–English bilinguals, etc.) and produced somewhat inconsistent transfer patterns, pointing to a need for further research (Leider et al. 2013; Proctor et al. 2012; Siu and Ho 2015, 2020; Swanson et al. 2008; Tong et al. 2021). From a theoretical perspective, the structural proximity between English and French syntax suggests that transfer may be expected across this language pair (Koda and Zehler 2008). Notably, Sohail et al. (2022) found direct and indirect relationships between English syntactic awareness and French reading comprehension in first-grade children who were English–French bilinguals. Building on Sohail et al. (2022), the third and fourth objectives of the present study are to examine the direct and indirect effects of English syntactic awareness on French reading comprehension in English–French bilingual children in Grade 2. We extend our previous work by examining bilingual children that are one year older than in Sohail et al. (2022) and by evaluating the mediating effects of French vocabulary in addition to French word reading. In doing so, we consider the factors that influence the patterns and direction of cross-language transfer among novice readers, broadening our theoretical understanding of transfer as it relates to L2 reading comprehension in the early elementary school period.

## 1.1. The Role of Syntactic Awareness in Reading Comprehension

Syntax is the set of rules that governs the arrangement of words and phrases to create sentences (Mackay et al. 2021). Syntactic awareness, as a metalinguistic skill, allows children to use their knowledge of syntax to understand and manipulate sentences (Cain 2007). Researchers theorize that syntactic awareness facilitates reading comprehension through sentence parsing, or chunking (Deacon and Kieffer 2018). Syntactic awareness enables children to break a complex sentence into its more manageable parts (e.g., phrases and clauses) (Mitchell 1994). Once the constituent parts are understood, they can be recombined to build a representation of the sentence and of the text (Perfetti and Stafura 2014). For example, if a child were to read the following sentence, 'For dinner, I like to eat pizza, but my brother prefers gnocchi', they may break it down into its dependent (For dinner) and independent clauses (I like to eat pizza//my brother prefers gnocchi). After processing each clause separately, the child can once again leverage awareness of syntax to reconstruct the sentence and determine the meaning of the entire sentence.

The direct within-language effect of syntactic awareness on reading comprehension has been substantiated by several studies involving English monolingual children (Deacon and Kieffer 2018; Mokhtari and Thompson 2006; Muter et al. 2004). This relationship has

also been found in French monolingual children (Demont and Gombert 1996; Gaux and Gombert 1999; Plaza and Cohen 2003; Plaza 2001). For example, Plaza and Cohen (2003) found that syntactic awareness predicted concurrent reading comprehension among first-grade French monolingual children after controlling for phonological processing, naming speed, word reading, and spelling.

To the best of our knowledge, few studies have examined the within-language relationship between syntactic awareness and reading comprehension in French among L2 children, and these studies produced mixed findings (Sohail et al. 2022; Lefrançois and Armand 2003; Simard et al. 2014). In a cross-sectional study involving Portuguese–French emergent bilinguals aged 10–11 years, which took place in Quebec, Canada, Simard et al. (2014) found that French syntactic awareness significantly contributed to French reading comprehension after controlling for age, receptive vocabulary, syntactic knowledge (measured through a syntactic preference task in which participants chose between two similar sentences) and phonological memory. Conversely, two studies involving French L2 learners found that syntactic awareness was correlated with reading comprehension in French among elementary-aged children but was not a unique predictor of reading comprehension outcomes (Lefrançois and Armand 2003; Sohail et al. 2022). Lefrançois and Armand (2003) assessed Spanish-speaking learners of French between 9 and 11 years of age who were recent arrivals in a predominantly French-speaking province of Canada. French syntactic awareness failed to explain unique variance in French reading comprehension after controlling for Spanish reading comprehension and French oral proficiency. It should be noted that controlling for Spanish reading comprehension may explain the lack of a significant relationship. More recently, in a study involving English–French bilinguals in French immersion in the first grade (ages 6–7), Sohail et al. (2022) did not find a significant relationship between French syntactic awareness and French reading comprehension after controlling for phonological awareness, receptive vocabulary, and non-verbal reasoning. Extending the developmental scope of our work, the present study examines this relationship among French immersion children in Grade 2.

## 1.2. The Indirect Relationship between Syntactic Awareness and Reading Comprehension as Mediated by Word Reading and Vocabulary

Scholars have proposed several mechanisms that might explain a within-language indirect relationship between syntactic awareness and reading comprehension through word reading (Bowey 1986; Tunmer et al. 1988). Tunmer et al. (1989) argued that beginning readers have limited decoding skills and, as a result, may be unable to accurately decode unfamiliar words encountered in text. Awareness of syntax may help readers use the constraints of sentential context to derive information about the word, such as its grammatical class, which can bolster partial decoding and enable word identification. In a similar vein, syntactic awareness may facilitate the decoding of irregular words, such as pint. The reader may partially decode the word and then use awareness of syntax to enable word identification based on syntactic context.

A small body of studies involving monolingual children has investigated whether syntactic awareness supports reading comprehension indirectly through word reading (e.g., Tunmer et al. 1989; Verhoeven and Perfetti 2008). In other words, these studies examined whether the within-language effect of syntactic awareness on reading comprehension is mediated through word reading. For example, in a longitudinal study following children from Grade 1 to Grade 2, Tunmer et al. (1988) found that the relationship between Grade 1 syntactic awareness and Grade 2 reading comprehension was mediated by phonological recoding (i.e., pseudoword reading) in Grade 1. In contrast, Deacon and Kieffer (2018) did not find evidence of mediation via word reading when they used Grade 3 variables (syntactic awareness, word reading) to predict reading comprehension assessed at Grade 4. The authors posit that the contrast with previous literature could be related to development. Younger students might rely on syntactic awareness to bolster underdeveloped skills in word reading and reading comprehension in turn. On the other hand, older students with

more fully developed word reading skills may leverage syntactic awareness to directly support reading comprehension.

Few studies have investigated the within-language indirect effect of syntactic awareness on reading comprehension through word reading in bilingual children. Tong et al. (2021) found that English syntactic awareness predicted English reading comprehension both directly and indirectly through word reading in a concurrent study of fourth-grade Chinese–English bilinguals. Whereas Sohail et al. (2022) found no evidence of a direct relationship between French syntactic awareness and French reading comprehension, they found that French word reading fully mediated the relationship between French syntactic awareness and French reading comprehension. The authors posited that emergent bilinguals rely heavily on word reading skills to comprehend L2 texts due to underdeveloped language skills.

There is an emerging body of literature investigating what role, if any, vocabulary may play in the relationship between syntactic awareness and reading comprehension. In children, 'syntactic bootstrapping,' is a mechanism that enables children to infer the semantic properties of novel words based on syntactic context (Babineau et al. 2020; Fisher et al. 2010, 2020; Naigles and Hoff-Ginsberg 1995; Naigles 1996). More specifically, children can use the syntactic context around an unknown word (i.e., the structure of a noun phrase: the boy wearing the turquoise shirt) to determine the grammatical category of an unknown word (i.e., adjective: turquoise) and, therefore, reduce the semantic category possibilities. A body of research has shown that there is a significant correlation between vocabulary skills and syntax in school-age children (e.g., Barbosa and Silva 2020; Mokhtari and Niederhauser 2012). To the best of our knowledge, Guo et al. (2011) conducted the only study exploring the mediating effect of vocabulary on the relationship between syntactic awareness and reading comprehension. The researchers showed that syntactic awareness predicted reading comprehension both directly and indirectly through vocabulary among monolingual English adults. The authors argued that even adult learners might leverage syntactic awareness to make use of contextual clues to acquire new and challenging vocabulary. However, the relationship between syntactic awareness, vocabulary, and reading comprehension has not been modeled in children. In the current study, we address the gaps in previous research by examining whether word reading and vocabulary might function as mediators between syntactic awareness and reading comprehension in emergent English–French bilingual children.

*1.3. Cross-Language Transfer of Syntactic Awareness to Reading Comprehension*

A number of theoretical frameworks have been proposed to account for the relationship between L1 and L2 language and literacy skills (e.g., Chung et al. 2019; Cummins 1979, 1981; Koda 2007; Koda and Zehler 2008). According to the Interdependence Hypothesis (Cummins 1979, 1981), L1 and L2 development are enabled by underlying proficiencies that are shared across languages. Koda (2007) proposed the Transfer Facilitation Model to elucidate the factors that may automatically activate the transfer of established L1 skills to developing L2 skills. The degree of structural similarity between the two languages and proficiency in the L1 are among these factors; both are of particular relevance to our study. However, the model was intended to account for transfer from the L1 to the L2 among older learners with established L1 skills. More recently, Chung et al. (2019) proposed the Interactive Transfer Framework to account more fully for the factors underlying transfer among emergent bilingual children. In addition to linguistic factors, this framework conceptualizes cross-language transfer as an interactive process that is influenced by sociolinguistic factors (e.g., age of acquisition, quantity and quality of L1–L2 input, context). The language learning context is particularly relevant in the current study as it targets children who are developing English and French skills simultaneously in French immersion programs.

With these frameworks in mind, it is important to contextualize the current study. French and English are both subject–verb–object (SVO) languages (Dekydtspotter et al. 1997; Lambrecht 1987). Thus, principal word order components (subject, verb, and object)

are ordered in the same way across both languages (Vinay and Darbelnet 1995). Other commonalities include the placement of auxiliary verbs (i.e., I am done/J'ai fini), direct and indirect objects (i.e., I gave the pencil to Marie/J'ai donné le crayon à Marie), and passive forms with and without agents (i.e., The book was destroyed by the children/Le livre a été détruit par les enfants). Word order differences between English and French are found in the placement of articles: adverbs (Marie often eats apples/Marie mange souvent des pommes), adjectives (The sad girl cries/la fille triste pleure), negation (We do not eat meat/On ne mange pas de viande), and pronouns (I gave it to him/Je le lui ai donné). We anticipate that the proximity between English and French word order might facilitate the transfer of syntactic awareness. Furthermore, students in French immersion are expected to have developed relatively strong English syntactic awareness through exposure and use in the home and community (Sohail et al. 2022). English syntactic awareness is, therefore, expected to transfer to French, the students' weaker language.

The cross-language transfer of syntactic awareness to reading comprehension has been studied among children from diverse backgrounds, including English–French (Sohail et al. 2022), Chinese–English (Siu and Ho 2015, 2020; Tong et al. 2021), and Spanish–English bilinguals (Leider et al. 2013; Proctor et al. 2012; Swanson et al. 2008). Sohail et al. (2022) found that English syntactic awareness exerted a direct effect on French reading comprehension after controlling for nonverbal reasoning, French phonological awareness, and French vocabulary among first-grade English–French emergent bilinguals. In addition, English syntactic awareness exerted indirect effects on French reading comprehension through French word reading and French syntactic awareness. As students had only two years of French exposure at the time of testing, the authors interpreted these findings to suggest that L1 skills support the development of L2 skills in the early years.

Consistent with Sohail et al.'s (2022) findings, two studies involving young Chinese–English bilinguals in Hong Kong reported the transfer of Chinese (L1) syntactic awareness to concurrent English (L2) reading comprehension (Siu and Ho 2015, 2020). In a study of first- and third-grade children, Siu and Ho (2015) tested children's ability to correctly order sentence segments in L1 Chinese. The authors found that Chinese syntactic awareness made a direct contribution to English reading comprehension, as well as an indirect contribution through English syntactic awareness, in both grades. They controlled for age, working memory, non-verbal intelligence, vocabulary, and word reading. In a subsequent study, Siu and Ho (2020) examined one-year longitudinal relationships between L1 syntactic awareness and L2 reading comprehension among Grade 1 and 3 Chinese–English bilingual children. They found that, for both age groups, performance on the word order task in Chinese at Time 1 predicted Time 2 English reading comprehension through Time 2 English syntactic awareness while controlling for the same variables. The authors speculated that the transfer of word order could be explained by the overlap in syntactic structure between Chinese and English.

Conversely, Tong et al. (2021) reported no evidence of cross-language transfer of syntactic awareness among Chinese–English bilinguals in the middle elementary years (i.e., ages 9–10), though this study showed that bilingual students performed better on items with shared features between the two languages as opposed to those with the unique features of each language. The authors posited that, as they age, children acquire sufficient skills in each of their languages to draw on within-language skills directly but may still use their knowledge of cross-language structures. Studies involving Spanish–English bilinguals in the United States have also failed to find evidence of the transfer of L1 syntactic awareness to L2 reading comprehension (Leider et al. 2013; Proctor et al. 2012; Swanson et al. 2008). For example, Proctor et al. (2012) followed students in Grades 2–4 for one academic year. Syntactic awareness, measured at the beginning of the study, was not found to predict either concurrent reading comprehension or growth in reading comprehension over time. The authors attributed their findings to the students' underdeveloped L1 skills relative to other bilingual samples. Given the heterogeneity of bilingual children, it is not surprising that previous research has produced somewhat inconsistent results. This inconsistency

highlights the importance of conducting additional research specific to a particular bilingual population and at different stages of development. Thus, the present study examined the cross-language transfer of syntactic awareness among English–French bilinguals in Grade 2 (i.e., one year older than the participants in Sohail et al. (2022)).

*1.4. The Present Study*

The purpose of the current study was to examine the contribution of syntactic awareness to reading comprehension in second-grade children who were emergent English–French bilinguals. Specifically, we analyzed the direct effects of French syntactic awareness on French reading comprehension, as well as the cross-language effects of English syntactic awareness on French reading comprehension. We also explored whether these within- and cross-language relationships were mediated by French word reading and French vocabulary. Specifically, we asked four research questions:

1.	Does French syntactic awareness have a direct effect on French reading comprehension?
2.	Does French syntactic awareness have indirect effects on French reading comprehension mediated through French word reading and vocabulary?
3.	Does English syntactic awareness directly affect French reading comprehension?
4.	Does English syntactic awareness have an indirect effect on French reading comprehension mediated through French word reading and French vocabulary?

The present study extends Sohail et al. (2022) by examining the contributions of English and French syntactic awareness to French reading comprehension in a different sample of emergent English–French bilingual children in Grade 2, an age group that has not been examined previously. For the first research question, Sohail et al. (2022) did not find a direct effect between French syntactic awareness and French reading comprehension. However, with an additional year of study in French, Grade 2 students may start to rely on within-language syntactic awareness. In other words, French syntactic awareness may predict French reading comprehension in our Grade 2 sample. For our second question, we expected to find the same mediation through word reading as in Sohail et al. (2022) because our participants were still in the early stages of learning French as an L2. Our study is the first to explore French vocabulary as a mediator. Mediation effects are expected given that children might use syntactic bootstrapping to learn new and challenging vocabulary (Babineau et al. 2020). For our third question, because English is still children's stronger language in Grade 2, we anticipate finding cross-language transfer from English syntactic awareness to French reading comprehension. Finally, for the fourth research question, we hypothesize that the cross-language relationship between English syntactic awareness and French reading comprehension is mediated through French word reading and French vocabulary, two skills that also mediate the within-language relationship.

The first and third questions concerning the direct effects of French and English syntactic awareness on French reading comprehension were examined using a hierarchical stepwise regression with French reading comprehension as the outcome variable. Non-verbal reasoning, working memory, French vocabulary, and French word reading were entered as control variables, and French and English syntactic awareness were entered as predictor variables. The second and fourth questions, which addressed the mediating effects of French word reading and vocabulary, were evaluated with two mediation models. The first model explored the relationship between French syntactic awareness and French reading comprehension as mediated by French word reading and French vocabulary. The second model explored the relationship between English syntactic awareness and French reading comprehension as mediated by French word reading and French vocabulary. Thus, the two models had either French or English syntactic awareness as the predictors, French word reading and vocabulary as mediators, and French reading comprehension as the outcome.

The participants of our study were enrolled in French immersion programs in Canada. French immersion programs are publicly funded programs offered to non-Francophone students throughout the country with a specific focus on French culture, language, and

literacy through content-based and action-oriented pedagogical approaches (Ministry of Education 2013). It is required that students achieve 3800 instructional hours by the end of Grade 8 (Ministry of Education 2013). In the school board where this study took place, for the first several years, teaching was conducted primarily in French (approximately 80%) with a select few subjects in English (such as music, drama, physical education, etc.). As children progress through the program, instruction in English increases and reaches about 50% by secondary school. Although students receive little or no English instruction in school in the early grades, they develop English language skills at home or in the community, as English is the dominant societal language (Au-Yeung et al. 2015).

## 2. Materials and Methods

### 2.1. Participants

A total of 68 second-grade children (40 females) were recruited from 7 French Immersion schools in a large metropolitan center of Canada. The average age of these students was 7–9 years (SD = 7 months). These children all learned French exclusively in school since senior kindergarten. On the other hand, they had minimal exposure to the French language outside the classroom, as English was the dominant societal language. Parents completed a demographic questionnaire adapted from the Alberta Language Environment Questionnaire (Paradis 2011), which provided information about parental education, language use, literacy activities at home, etc. Among the 68 children, 62 spoke English as the L1, and 6 students were English Language Learners (ELLs). A student was considered ELL if they were exposed to another language other than English at least 50% of the time at home and if they spoke a language other than English as their first language. These 6 home languages represented were Cantonese, Yomba, Arabic, Gujarati, Tamil, and Vietnamese. The average level of education between both guardians of children was at the college/university level.

### 2.2. Measures

A battery of measures was administered to all students in the spring of Grade 2, including French reading comprehension, English and French syntactic awareness, French word reading, French receptive vocabulary, non-verbal reasoning in English, and working memory in English. The ALEQ demographic questionnaire was filled out by parents in English.

### 2.2.1. French Reading Comprehension

Level B, Form 4 of the Gates–MacGinitie Canadian reading comprehension test, second edition (GMRT-II; Gates and MacGinitie 1992) was translated into French by a native speaker of the language. This translated French reading comprehension measure has been used in several previous studies (e.g., Lam et al. 2020; Lee and Chen 2019; Hipfner-Boucher et al. 2016). In this test, children were required to read passages and answer multiple-choice questions. In total, the test included 12 passages and 46 multiple-choice items. Children were given 30 min to complete the test. Children completed the test in small groups supervised by a trained research assistant.

### 2.2.2. English and French Syntactic Awareness

An experimental measure was used to assess word order correction in English and French. This task was developed based on similar word order correction tasks (e.g., Deacon and Kieffer 2018). Both versions of the task contained 20 items as well as 2 practice items. For the tasks in both languages, students were presented with a sentence containing syntactic errors and asked to correct the order of words in a sentence. As an example, given "the meowed black cat", the corrected response was "the black cat meowed". These tasks included syntactic structures that are commonly found in each respective language. Please see Table A1 (Appendix A) for examples from each language.

### 2.2.3. French Receptive Vocabulary

To assess receptive vocabulary, the Échelle de Vocabulaire en Images Peabody Form A (EVIP; Dunn et al. 1993) was administered. This test involved a stimulus booklet composed of 170 items. Each item consisted of four pictures. Children were aurally given a word and asked to either point to or name the number of the corresponding picture on a page. Since the basal start points were standardized using Francophone children, we started all children at the first item regardless of their age to prevent floor effects. This test was administered using standardized procedures. The test was discontinued when children inaccurately identified six pictures in a set of eight.

### 2.2.4. French Word Reading

To assess French word reading, the Lecture de Mots (word reading) subtest of the Wechsler Individual Achievement Test-Second Edition (WIAT-II; Wechsler 2005) was utilized. This measure has been standardized on a sample of Canadian Francophone children. It was composed of 84 items, requiring children to read aloud words of increasing difficulty. The test was discontinued when children inaccurately identified seven words in a row.

### 2.2.5. Non-Verbal Reasoning

The Matrix Analogies Test (MAT; Naglieri 1985) was used to assess non-verbal reasoning. Due to time constraints, only the reasoning by analogy subtest was used. Each item included a pattern of pictures and a question mark. The child was asked to choose from the five or six pictures at the bottom to complete the pattern. There were 16 items in total, and testing was discontinued once a child received a 0 on 4 consecutive items.

### 2.2.6. Working Memory

The Memory for Digits subtest of the Comprehensive Test of Phonological Processing (CTOPP-II; Wagner et al. 2013) was used to assess working memory. In this test, children would hear number sequences that increased in length as the test continued, from two numbers to nine numbers. Students would only hear the items once and would receive one point for each item completed without any errors (e.g., omission, switching, code-switching). There were 28 items total, including 4 practice items where students received feedback and correction if necessary. Testing was discontinued after a student made an error on three consecutive items.

### 2.2.7. Demographic Questionnaire

A shortened version of the Alberta Language Environment Questionnaire was used (ALEQ; Paradis 2011). In this questionnaire, parents and/or guardians answered questions about themselves (i.e., parent education and income) as well as their children (i.e., age, gender, languages spoken at home, etc.).

### 2.3. Procedure

Children were tested in their schools by an extensively trained team of graduate and undergraduate students. All testers were highly proficient in English and French. Students completed testing with a research assistant in a quiet environment outside the classroom for most measures. Reading comprehension was completed in small groups, whereas all other measures were completed individually. Students were assessed during several 1 h sessions (4–6 depending on the child) in which they were given breaks and asked for verbal assent throughout the session.

## 3. Results

### 3.1. Descriptive Statistics

Table 1 presents the ranges, means, standard deviations, and reliability measures (Cronbach's $\alpha$) for the seven measures included in this analysis: working memory, non-verbal reasoning, French receptive vocabulary, French word reading, English syntactic

awareness, French syntactic awareness, and French reading comprehension. The skewness and kurtosis values of all measures are displayed in Table 1. The standard error for skewness was 0.291, and the standard error for kurtosis was 0.574. With the exception of French syntactic awareness, all measures in this study were normally distributed. The kurtosis for French syntactic awareness was 3.718, which exceeded the +2-criterion proposed by Tabachnick et al. (2007). The French syntactic awareness measure was, therefore, transformed using the log function to correct for the skew. As some students had an overall score of zero, the Lg10 (maximum value +1) was implemented. Once transformed, the transformed variable was compared to the original variable in all analyses. No significant differences were found between the two variables, and, therefore, the original French syntactic awareness variable was used in all subsequent analyses. A paired-sample *t*-test found that students performed significantly better in English syntactic awareness than in French syntactic awareness, ($t(67) = -17.596$, $p < 0.001$), suggesting that English was the children's stronger language.

**Table 1.** Descriptive statistics of all measures ($n = 68$).

| Measures | Range | *M* | *SD* | Cronbach's $\alpha$ | Skewness | Kurtosis |
|---|---|---|---|---|---|---|
| 1 NVR | 0–13 | 4.46 | 3.483 | 0.914 | 0.888 | −0.154 |
| 2 WM | 11–22 | 15.82 | 2.259 | 0.932 | 0.498 | −0.195 |
| 3 FRVC | 11–95 | 46.90 | 19.458 | 0.954 | 0.529 | −0.211 |
| 4 FRWR | 0–71 | 32.12 | 19.777 | 0.978 | 0.216 | −1.104 |
| 5 ENSA | 1–18 | 9.25 | 4.262 | 0.754 | −0.097 | −1.046 |
| 6 FRSA | 0–11 | 2.19 | 2.326 | 0.772 | 1.763 * | 3.718 * |
| 7 FRRC | 1–36 | 20.56 | 7.39 | 0.959 | 0.030 | −0.121 |

NVR = non-verbal reasoning; WM = working memory; FRVC = French receptive vocabulary; FRWR = French word reading; ENSA = English syntactic awareness; FRSA = French syntactic awareness; FRRC = French reading comprehension. * $p < 0.05$.

Six moderate univariate outliers were found across five measures: English syntactic awareness, French syntactic awareness, French word reading, French vocabulary, and French reading comprehension. However, a comparison of results both including and excluding outliers revealed no significant changes in significance patterns. Furthermore, as discussed by Anscombe (1960), since the population variance was known to the research team, we chose to keep these outliers in the analysis to show a more representative distribution. Using Malahanobis distance (Stevens 1984), the data were also checked for multivariate outliers, which were not found.

### 3.2. Correlations

Most of the measures were significantly correlated (Table 2). French syntactic awareness was significantly correlated with French reading comprehension ($r = 0.41$, $p < 0.01$), French word reading ($r = 0.47$, $p < 0.01$), and French receptive vocabulary ($r = 0.33$, $p < 0.01$). English syntactic awareness was significantly correlated with French reading comprehension ($r = 0.54$, $p < 0.01$), French word reading ($r = 0.47$, $p < 0.01$), and French receptive vocabulary ($r = 0.41$, $p < 0.01$). Finally, syntactic awareness in French and English were correlated ($r = 0.42$, $p < 0.01$).

**Table 2.** Correlations between all measures (*n* = 68).

| Measures | 1 | 2 | 3 | 4 | 5 | 6 |
|----------|-----|-----|-----|-----|-----|-----|
| 1 NVR | - | | | | | |
| 2 WM | 0.100 | - | | | | |
| 3 FRVC | 0.053 | 0.125 | - | | | |
| 4 FRWR | 0.206 | 0.166 | 0.306 * | - | | |
| 5 FRSA | 0.049 | 0.116 | 0.334 ** | 0.465 ** | - | |
| 6 ENSA | 0.091 | 0.100 | 0.410 ** | 0.473 ** | 0.418 ** | - |
| 7 FRRC | 0.203 | 0.247 * | 0.419 ** | 0.679 ** | 0.405 ** | 0.541 ** |

NVR = non-verbal reasoning; WM = working memory; FRVC = French receptive vocabulary; FRWR = French word reading; ENSA = English syntactic awareness; FRSA = French syntactic awareness; FRRC = French reading comprehension. ** $p < 0.01$; * $p < 0.05$.

### 3.3. Hierarchical Stepwise Regression Models

Hierarchical stepwise regression was performed to predict French reading comprehension. Given the relatively small sample size of this study, the stepwise regression method was used in all steps in order to reduce the number of predictors. This method has been used in previous papers (e.g., Perfetti and Liu 2005; Wang et al. 2006). In step one, the cognitive variables were added: non-verbal reasoning (NVR) and working memory (WM). The stepwise method removed NVR, leaving WM as a significant control factor, $F_{(2, 64)} = 0.061$, $p < 0.05$. The next step was to add receptive vocabulary (FRVC) and word reading (FRWR). The stepwise method kept FRWR in the model, $F_{(2, 64)} = 0.419$, $p < 0.001$, as well as FRVC, $F_{(2, 64)} = 0.045$, $p < 0.05$. French and English syntactic awareness, as the final predictor variables, were entered in step three (FRSA, French; ENSA, English). Only English syntactic awareness remained in the model after the stepwise method, $F_{(2, 64)} = 0.030$, $p < 0.05$ As can be seen in Table 3, in total, 56% of the variance in French reading comprehension is explained by this model. This model was also run without English syntactic awareness in the last step in order to determine the independent contribution of French syntactic awareness. However, even without English in the model, French syntactic awareness was still removed in the final step.

**Table 3.** Hierarchical stepwise regression analyses examining the role of receptive vocabulary, word reading, and syntactic awareness in French reading comprehension (*n* = 68).

| Predictors | $R^2$ | $\Delta R^2$ | $\Delta F$ | β | t |
|------------|---------|-----------|-------|-------|---------|
| Step 1 | | | | | |
| WM | 0.061 | 0.061 * | 4.303 | 0.247 | 2.074 * |
| NVR | Removed | | | | |
| Step 2 | | | | | |
| FRWR | 0.480 | 0.419 *** | 52.299 | 0.656 | 7.232 *** |
| FRVC | 0.525 | 0.045 * | 6.049 | 0.223 | 2.459 * |
| Step 3 | | | | | |
| FRSA | Removed | | | | |
| ENSA | 0.555 | 0.030 * | 4.219 | 0.209 | 2.054 * |

WM = working memory; NVR = non-verbal reasoning; FRWR = French word reading; FRVC = French receptive vocabulary; FRSA = French syntactic awareness; ENSA = English syntactic awareness. *** $p < 0.001$; * $p < 0.05$.

### 3.4. Mediation Modelling

To examine the effects of syntactic awareness both within and between languages, PROCESS (Hayes 2017) was used to model the relationships between English and French syntactic awareness and reading comprehension in French. Due to the small sample size included in this paper, only the outcome, mediating, and target variables were included in these models. Control variables, which were included in the hierarchical stepwise regression above, were not included. The details of these models are discussed below.

The first model, which is a within-language model, explored the relationship between French syntactic awareness and French reading comprehension as mediated by French word

reading and French vocabulary. As shown in Figure 1, an examination of the proposed within-French model showed several interesting results. The indirect effect of French syntactic awareness on French reading comprehension through French word reading was statistically significant (effect = 0.8154, 95% C.I. (0.2713, 0.3844)). The indirect effect of French syntactic awareness on French reading comprehension through French vocabulary was statistically significant (effect = 0.2083, 95% C.I. (0.1125, 0.0419)). However, the direct effect of French syntactic awareness on French reading comprehension was not significant (effect = 0.1332, $p$ = 0.68, 95% C.I. (−0.5110, 0.7774)). Therefore, the relationship between French syntactic awareness and French reading comprehension was fully mediated by French word reading and French vocabulary.

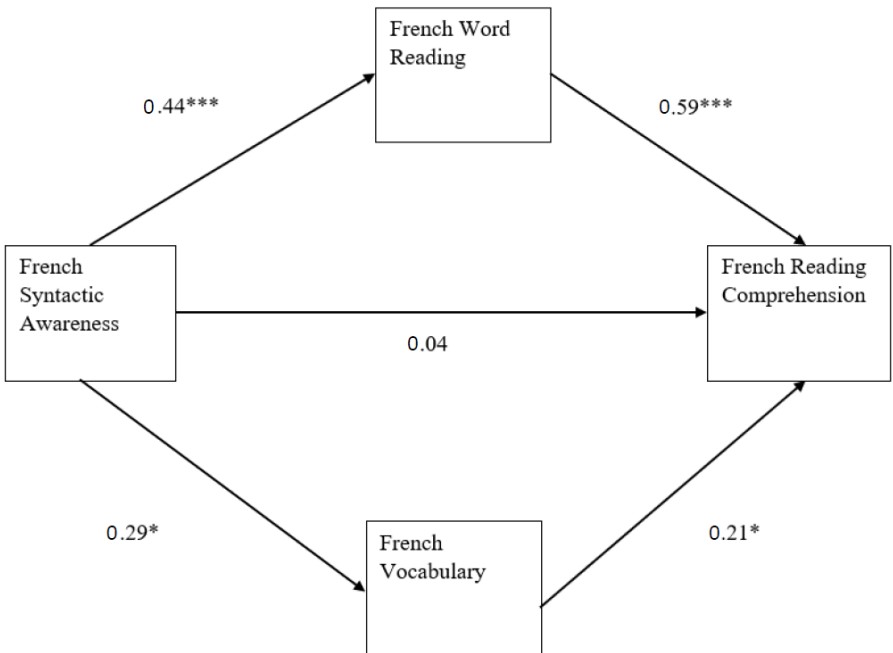

**Figure 1.** The French within-language model with standardized coefficients. *** $p < 0.001$; * $p < 0.05$. All coefficients are standardized.

The second model, which is a cross-language model, explored the relationship between English syntactic awareness and French reading comprehension as mediated by French word reading and French vocabulary. Since French syntactic awareness did not predict French reading comprehension in the regression analysis, it was not included as a mediator in the cross-language model. As shown in Figure 2, an examination of the proposed cross-language model showed several interesting results. The indirect effect of English syntactic awareness on French reading comprehension through French word reading was statistically significant (effect = 0.3771, 95% C.I. (0.1206, 0.1605)). The indirect effect of English syntactic awareness on French reading comprehension through French vocabulary was not significant (effect = 0.1187, 95% C.I. (−0.0023, 0.2699)). Finally, the direct effect of English syntactic awareness on French reading comprehension was statistically significant (effect = 0.3933, $p < 0.05$, 95% C.I. (0.0443, 0.7422)). Therefore, the relationship between English syntactic awareness and French reading comprehension was partially mediated by French word reading but not by French vocabulary.

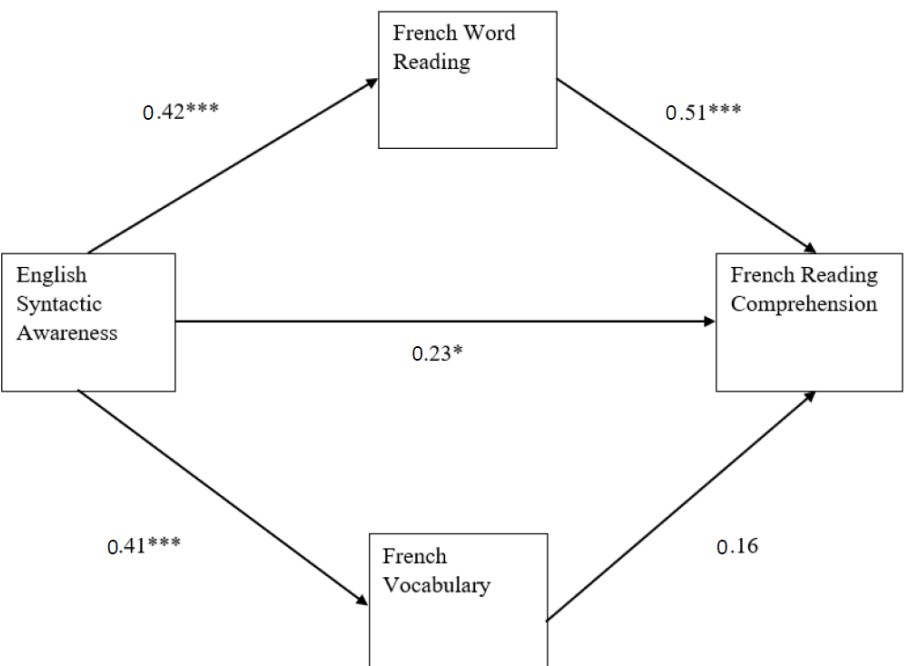

**Figure 2.** The cross-language model with standardized coefficients. *** *p* < 0.001; * *p* < 0.05. All coefficients are standardized.

## 4. Discussion

The purpose of the current study was to investigate the direct and indirect relationships between both French and English syntactic awareness and French reading comprehension in emergent English–French bilingual children. We did not observe a direct effect of French syntactic awareness on French reading comprehension. However, French syntactic awareness had indirect effects on French reading comprehension mediated through French word reading and French receptive vocabulary. With respect to the cross-language results, we observed a direct effect of English syntactic awareness on French reading comprehension. We also observed an indirect effect from English syntactic awareness to French reading comprehension mediated through French word reading. Mediation through French vocabulary, however, was not significant in the cross-language model. These findings and their implications are discussed in more detail below.

### 4.1. The Role of Syntactic Awareness in Reading Comprehension

Our first question concerned the direct relationship between French syntactic awareness and French reading comprehension. The results showed that French syntactic awareness was not a significant predictor of French reading comprehension in English–French emergent bilinguals in Grade 2. This might be explained by the children's relatively low performance on the French syntactic awareness task. Given that our participants lived in an English-dominant environment and had only had French instruction for less than three years, it is possible that their French syntactic awareness was not sufficiently developed to support French reading comprehension. In French monolingual populations, a direct relationship was found between French syntactic awareness and French reading comprehension in children as young as the first grade (Plaza and Cohen 2003). In contrast, this relationship was only observed in the fifth grade (Simard et al. 2014), but not earlier (e.g., Sohail et al. 2022; Hermanto et al. 2012), in bilingual populations. Thus, it may take longer for syntactic awareness to emerge as a significant predictor of reading comprehension among French L2 learners. Notably, French word reading explained the most unique variance in our regression and mediation models, confirming that children rely primarily on decoding for text comprehension in the initial stages of reading development (Hoover and Gough 1990; Lonigan et al. 2018; Verhoeven and Leeuwe 2012).

*4.2. The Indirect Relationship between Syntactic Awareness and Reading Comprehension as Mediated by Word Reading and Vocabulary*

Our second question explored the indirect effects of French syntactic awareness on French reading comprehension as mediated through French word reading and French receptive vocabulary. While the direct pathway was not significant, the relationship between French syntactic awareness and French reading comprehension was mediated by both French word reading and French receptive vocabulary. Our finding with respect to the indirect effect of word reading on reading comprehension aligns with previous studies (Sohail et al. 2022; Tong et al. 2021) involving bilingual children in the early primary grades, suggesting that syntactic awareness is activated in support of word reading in the L2. In concrete terms, decoding new and challenging words is accomplished by a combination of partial decoding and an awareness of the surrounding syntax to narrow the grammatical category of the word. In monolingual English populations, word reading supports the relationship between syntactic awareness and reading comprehension in the early grades (e.g., Tunmer et al. 1988), but is no longer a significant support by the fourth grade (Deacon and Kieffer 2018). Future studies should investigate whether this pattern will also emerge among bilingual students.

In addition to French word reading, French receptive vocabulary was found to mediate the relationship between French syntactic awareness and French reading comprehension in the within-language model. As shown in Babineau et al. (2020), syntactic awareness might bolster vocabulary learning through syntactic bootstrapping in preschool children. The results of the current study suggest that syntactic bootstrapping might also support vocabulary learning in school-age children. In this vein, syntactic awareness helps children acquire challenging words when they read for meaning (Fisher et al. 2010, 2020; Naigles and Hoff-Ginsberg 1995; Naigles 1996). This finding was corroborated in a study involving adults, in which vocabulary mediated the relationship between syntactic awareness and reading comprehension (Guo et al. 2011). In this study specifically, vocabulary was assessed using new and challenging words for adult-level readers. The current study, in conjunction with the previous findings, suggests that syntactic awareness supports novel vocabulary learning from preschool well into adulthood. While it was originally proposed in relation to oral language acquisition among young children, syntactic bootstrapping may extend into word acquisition in written language and in turn facilitate reading comprehension among older children and adults.

*4.3. Cross-Language Transfer of Syntactic Awareness to Reading Comprehension*

Our third question addressed cross-linguistic effects, investigating the direct effect of English syntactic awareness on French reading comprehension. In contrast to the French within-language model, there was a unique and significant direct relationship between English syntactic awareness and French reading comprehension. This study is consistent with Sohail et al. (2022), who found a similar direct effect in the first grade. Thus, in the initial stages of literacy development, emergent English–French bilingual children can use syntactic awareness developed in English to bolster reading comprehension in the L2. The structural similarity between English and French, as SVO languages, may be one of the key reasons for transfer (Koda 2007; Koda and Zehler 2008). Notably, syntactic awareness has also been found to transfer from Chinese to English, both of which are SVO languages (Siu and Ho 2020; Tong et al. 2021). Future studies need to explore whether syntactic awareness also transfers between children whose two languages share fewer syntactic features. In terms of directionality, we observed a transfer from English to French because our participants were more proficient in their L1 (Koda 2007; Koda and Zehler 2008). In addition to these linguistic factors, the sociolinguistic context would have further facilitated a transfer from English to French, as French Immersion is an additive bilingual program (Au-Yeung et al. 2015).

Our fourth and final question, which addressed the indirect effect of English syntactic awareness on French reading comprehension through French word reading and French

receptive vocabulary, yielded more nuanced results. English syntactic awareness had an indirect effect on French reading comprehension mediated through French word reading but not through French vocabulary. This cross-language finding mirrors the within-language model in which word reading was also found to be a significant mediator. These findings further extend the results of Sohail et al. (2022), who found the same relationship in younger students. Given that students are still beginning readers of French in Grade 2, their reliance on English syntactic awareness to facilitate French word decoding fits with their current developmental stage. On the other hand, unlike the within-language results, the mediational pathway through French vocabulary was not significant in the cross-language model. Since French vocabulary was already a weak mediator in the within-language model, its mediating effect was further attenuated in the cross-language model in which English syntactic awareness was a strong direct contributor to French reading comprehension. We furthermore suspect that the strong facilitation from English syntactic awareness made the contribution of vocabulary redundant in the cross-language model.

*4.4. Theoretical and Educational Implications*

A significant body of research attests to the influence that metalinguistic skills have on learning to read both within and across languages, with the bulk of this research focusing on phonological and morphological awareness (for a review, see Luo et al. 2014; Kuo and Anderson 2008). In the current work, however, we examined the effect of a less-studied dimension of metalinguistic awareness—syntactic awareness—assessed in both the L1 (English) and L2 (French) on L2 reading comprehension. We found evidence that, among novice L2 readers, syntactic awareness does indeed exert an influence on reading comprehension, through both direct and indirect effects. Our study thus contributes to the small body of research that has reported the facilitative effects of metalinguistic skill on reading comprehension beyond the well-documented effects of phonological and morphological awareness. Moreover, the use of mediational modeling techniques allowed us to reveal the complex patterns of influence L1 and L2 syntactic awareness exerts on L2 reading comprehension among young bilingual readers, advancing our understanding of the processes that underlie it.

Importantly, the current study also enhances our understanding of the factors that may influence the patterns and direction of cross-language transfer specifically among novice readers in FI. Our finding of a direct effect of English syntactic awareness on French reading comprehension, particularly in the absence of a parallel within-language effect, suggests that relative L1–L2 proficiency may play a role in enabling the transfer of skills. Indeed, we posit that, in the earliest stages of learning, children draw on skills developed in their stronger language to support additional language learning. Among early elementary-aged children in FI, that language is English. However, it is important to keep in mind that the children in our study had yet to begin formal English language arts instruction. It would appear, then, that the level of syntactic awareness acquired in the L1 during the preschool and early elementary years through English language exposure and use was sufficient to support transfer to reading comprehension across languages. Consequently, our study provides evidence of the important role played by sociolinguistic factors (e.g., the value placed on English–French bilingualism within the Canadian context, the promotion of additive English–French bilingualism in FI programs), in addition to linguistic factors (language proximity, proficiency), in enabling transfer. In this respect, our results align with the *Interactive Transfer Framework*, which proposes that the language learning environment influences the pattern and direction of transfer in young bilingual learners (Chung et al. 2019).

At the same time, we report findings that attest to the impact of the learning context in supporting L2 reading comprehension among FI learners. Specifically, instructional factors may explain, to some extent, our within-language results, which revealed only indirect effects of French syntactic awareness on reading comprehension via word reading and vocabulary. Whereas word reading and vocabulary are the focus of direct instruction in FI classrooms in the early elementary grades, explicit instruction in French grammar is not

mandated (Ministry of Education 2013). As a result, the French syntactic awareness skills of FI learners may remain underdeveloped relative to their word reading skills and vocabulary. When reading for meaning in the L2, therefore, FI children in Grade 2 are largely reliant on those L2 skills that have been the subject of direct instruction. We speculate that instruction to promote awareness of L2 sentence structure may benefit L2 reading comprehension. Such instruction would ideally be embedded in authentic learning tasks that support the communicative goals of French immersion. Importantly, a limited number of studies have demonstrated the effectiveness of syntactic interventions in promoting syntactic awareness among monolingual children (see Mackay et al. 2021, for a review).

While this study makes novel contributions, it also has a few limitations. First, due to the small sample size, we were not able to control for other aspects of metalinguistic awareness, such as morphological awareness, in our regressions. However, several previous studies have shown that syntactic awareness was a unique predictor of reading comprehension even after controlling for morphological awareness (i.e., Deacon and Kieffer 2018; Tong et al. 2021). Furthermore, we did not include control variables in the mediation models. As such, the mediation results were preliminary and should be interpreted with caution. Future studies with larger sample sizes will be able to use structural equation modeling (SEM) to analyze the direct and indirect effects of syntactic awareness on reading comprehension after accounting for control variables. The SEM analyses will also provide model fit indexes. In terms of measures, for French reading comprehension, a translated version of Gates–MacGinitie was used (GMRT-II; Gates and MacGinitie 1992). This decision was made due to a dearth of French reading comprehension measures developed specifically for French immersion students. A further limitation of this study is the relatively low performance in the French syntactic awareness measure due to the children's limited French proficiency. To account for these varying levels of proficiency across languages, future studies should plan to include more developmentally appropriate items in the L2 (i.e., simple sentences only in Grade 2 and younger).

There are three components of this study that especially warrant further exploration. The first is that the results of this study are contingent upon the age and development of the children. While English syntactic awareness is a strong predictor of French reading comprehension at this age, French syntactic awareness may play a larger role as children develop a higher level of French proficiency. At the same time, we might see a reduced mediating role for French word reading as children increasingly rely on French vocabulary for French reading comprehension. Thus, future studies should evaluate the levels of syntactic awareness, as well as its effects on reading comprehension in older English–French bilingual children. Given that our study only had a small number of ELL children, we were not able to analyze them as a separate group. Further studies would also benefit from recruiting more ELL students in French immersion programs in order to compare the patterns of the results between English L1 and ELL students. It would be interesting to extend our study to language pairings other than English–French (i.e., Spanish–English and Arabic–English) to explore how the transfer of syntactic awareness is influenced by the linguistic features of the L1 and L2. Finally, future research examining the impact of relative L1–L2 proficiency over time on the cross-language transfer of skills among children in FI is clearly warranted. Ideally, future studies would assess both L1 and L2 reading comprehension. It may be the case that, as children's level of proficiency in French increases over time, evidence of the within-language transfer of syntactic awareness to L2 reading comprehension would emerge as it did in the study by Simard et al. (2014). Alternatively, we cannot exclude the possibility that, with increased proficiency in French, evidence of back-transfer (from the L2 to the L1) may emerge as it did in Deacon et al. (2007) in relation to morphological awareness.

## 5. Conclusions

In summary, the current study examined the contributions of French and English syntactic awareness to French reading comprehension. We found that French syntactic

awareness did not directly support French reading comprehension in Grade 2; its effects were instead fully mediated through French word reading and French vocabulary, suggesting that French syntactic awareness supports both the partial decoding of novel words through context, as well as the acquisition of challenging vocabulary through syntactic bootstrapping. Furthermore, English syntactic awareness supported French reading comprehension both directly and indirectly through French word reading. In line with several cross-language transfer theories (e.g., Chung et al. 2019; Cummins 1979, 1981; Koda 2007), these findings are consistent with the cross-language direct effect, in that transfer may be influenced by the relative levels of L1 and L2 proficiency and structural similarity, as well as the language immersion context. With respect to educational implications, it may be helpful to build on the strengths that children have developed in the L1 to support L2 learning. Specifically, children's understanding of similarities (e.g., the SVO sentence structure) and differences (e.g., placement of adjectives) in syntactic structure between English and French could enable English–French bilinguals to build more syntactically accurate sentences in their L2.

**Author Contributions:** Conceptualization, X.C. and S.H.D.; methodology, X.C. and S.H.D.; formal analysis, P.W.K., D.B. and X.C; data curation, D.B. and X.C.; writing—original draft preparation, D.B., K.H.-B. and X.C; writing—review and editing, D.B., K.H.-B., P.W.K., X.C. and S.H.D.; visualization, D.B.; supervision, X.C. and S.H.D.; project administration, X.C. and S.H.D. funding acquisition, X.C. and S.H.D. All authors have read and agreed to the published version of the manuscript.

**Funding:** This research was funded by SSHRC, grant number 494811 and the APC was funded by SSHRC.

**Institutional Review Board Statement:** The study was conducted in accordance with the Declaration of Helsinki and approved by the Institutional Review Board of the University of Toronto (30636, 1 April 2016).

**Informed Consent Statement:** Informed consent was obtained from all subjects involved in the study.

**Data Availability Statement:** The data presented in this study are available on request from the corresponding author. The data are not publicly available due to the age of the participants.

**Conflicts of Interest:** The authors declare no conflict of interest.

## Appendix A

**Table A1.** Sample items from the syntactic awareness measure.

| Language | Test Item | Correct Response |
| --- | --- | --- |
| English | The dog, by, were eaten, the cookies. | The cookies were eaten by the dog. |
| English | In her notebook, Jane, it, draws. | Jane draws it in her notebook. |
| French | Tombent, branche, de la, les feuilles. | Les feuilles tombent de la branche. |
| French | Pleure, le, triste, garçon. | Le garçon triste pleure. |

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
