# Peer review of "Syntactic Awareness and Reading Comprehension in Emergent Bilingual Children"

_languages, doi:10.3390/languages8010062_

Round 1

Reviewer 1 Report

Overall Impressions

Thank you for the opportunity to review this well-written manuscript entitled “Syntactic Awareness and Reading Comprehension in Emergent Bilinguals”. The article reported the role of L1 and L2 syntactic awareness on L2 reading comprehension in a sample of Grade 2 French immersion students. Where L2 syntactic awareness was completely mediated by L2 word reading and L2 vocabulary, L1 syntactic awareness had both direct effects and indirect effects through L2 word reading. Since syntactic awareness is often less emphasized than word reading and vocabulary in L2 reading comprehension, this study makes an important contribution to our knowledge. I do have some suggestions that I think would strengthen the manuscript.

Introduction  

Overall, the introduction is well-written and motivates the study nicely. However, it would be helpful if the authors explained earlier why looking at similar research questions in Grade 2 to their earlier work in Grade 1 is important. This information does materialize at the end of the introduction, but there are a few places earlier where the authors indicate they are extending their work, without indicating why a single grade difference is so important and why they might expect different results.

Participants

 Could you please clarify the format of the French immersion program that these students are enrolled in? There are subtle differences between school boards and it would be important to know the nature of their exposure to both French and English in their context.

Could you indicate whether these participants are the same participants as reported in AUTHOR (2022). If so, please discuss the implications of using the same sample.

Method and Results

As is nicely pointed out, participants performed poorly on the French syntactic measure, and it is unfortunate that this measure is central to the research questions provided. It seems that the mode is likely zero on this test and is not doing a great job teasing participants’ abilities apart. I wonder if the authors explored other more lenient scoring methods than simply “correct” or “incorrect”. For example, if some words were slotted into the correct part of the sentence could partial credit be given?

I also wonder if a sentence could be added about whether French Syntactic awareness was a significant unique predictor if English Syntactic awareness was not included in the regression model at the last step? My concern is that English was a stronger predictor because the measure was better (i.e., more age appropriate) rather than because of the underlying skill.

Figure 1 is really a table and likely could be an appendix. Of note, the document says there is an appendix and then one is not present. I also wonder about the emphasis on items that have similar or different structure across languages. This distinction is not relevant in any of the analyses, so it is not clear why it is being highlighted here.

p. 9, line 419, it wasn’t clear to me why French syntactic awareness was removed from the cross-language mediation analysis but remained in the within language analysis.

 Discussion

The discussion is well-written, does a good job addressing each research question and highlighting key limitations/directions for future research. However, there is only a sentence or two about the implications for education. I would like to see more elaboration on the implications for literacy instruction in a bilingual context. This is particularly relevant given the recent release of the right to read report in Ontario Canada. How does the data speak to what skills are important to foster in developing reading comprehension ability? What are the data’s limitations to do so?  

 Minor comments

If Cronbach’s alpha is reported in Table 1, I don’t think it needs to be reported with each of the measures.

p. 6, line 271, do you have a more recent reference that Swain (2000)? Perhaps from a government source?

p. 6, line 279, what does “s-grade” mean?

In Table 1, could you change the “min-max” to “range” and then perhaps provide the max score the participant could have received in a separate column?

p. 8, line 379. The brackets are missing around the degrees of freedom and the p-value should be less than 001 rather than .000.

For all tables, please be consistent in the naming of the measures. Some variables are named based on their test (e.g, EVIP) and others based on what is being measured (NVR).

Author Response

Hello,

Thank you so much for your insightful comments on our manuscript. Please find our response letter attached. We will also be uploading a revised copy of the manuscript.

Reviewer 2 Report

Manuscript number – languages-1844676

Title -  Syntactic Awareness and Reading Comprehension in Emergent Bilingual Children

This study, as part of a special issue on second language reading acquisition in languages with different writing systems, takes up the question of direct and indirect influences of syntactic awareness (SA) (within language involving L2 SA, and between-language involving L1 SA) on L2 reading comprehension (RC) in young children with emerging spoken and written language proficiency in the L2. Using a sample of 68 English-L1 children learning French (the L2) in French-Immersion contexts, the authors reported significant within-language indirect (but not direct) influence of French SA on French RC, as well as direct (and partly indirect via French word reading) influence of English SA on French RC. The study was well designed and executed, results were analyzed with appropriate tests (using stepwise multiple regression and PROCESS analyses to identify direct and indirect influences), and conclusions were good reflections of the patterns identified.

The study arrived in type-set form, and so I assume that it has already been reviewed somewhat by editors. My impression is that the study and the write up are polished and well-done, with very few revisions necessary. It's not clear, however, how this paper fits in with the special-topics issue on reading in different writing systems. Given that this study is about alphabetic writing systems, perhaps some context should be provided to set the stage for this manuscript showing how this study fits within the special topics issue, unless this will be done in the editor’s introduction to the special issue.

Some (mostly minor) recommendations and comments:

The theoretical treatment, especially around the meaning of indirect effects, was very well done and sophisticated. I would like to see a bit more elaboration, though, on what the patterns of results can mean for current perspectives on cross-language facilitation and/or inhibition for reading comprehension in a weaker language.

Proficiency in the L2 was an important factor, and in a sense a moving target given that a very wide range of individual differences in L2 proficiency are likely to be present in the sample (given the range of scores reported on the French SA – the skewness is consistent with a bottoming-out of scores on that measure and make the outcome difficult to interpret; good discussion though of this as a limitation and how it could be ameliorated in the future to obtain more-valid indication of true ability on this factor - and French vocabulary). It would be a good idea to elaborate a bit more on how different levels of proficiency in an L2 could result in varied patterns involving direct and indirect influences of French SA on French reading comprehension – either in the Introduction in the context of other cross-language literature reviewed (especially in describing any information provided in those studies about the roles of L2 proficiency), or in the Discussion in the interpretation of obtained results. Also, some discussion of individual differences in L1 word reading, vocabulary, and reading comprehension could provide important contextual information for the reader to understand these characteristics of the sample – were there any struggling readers in L1, for example, or children with low cognitive ability, given the range of NVR scores reported.

At the outset, the authors should give more details on why Grade 2 children were sampled and what this could imply for the patterns of results that were expected. At the end of the third paragraph of section 1.2 (which provides an excellent discussion of indirect effects), there’s a bit of ambiguity in the wording – in the second-last sentence starting with “Whereas”, is “mediated” being used interchangeably with “Indirect effect”? This seems to be what’s implied given the context of the paragraph, and if so it would be best to use consistent wording for this –refer to “indirect effect” throughout if this is what’s intended. Also, was there any measure of L2 proficiency carried out in that study, and if so was it included in the analyses? Although this may be outside the scope of the authors’ intent, I wonder if it would make sense to mention relevant literature on the contributions of morphological awareness (in addition to SA and Vocab) on L2 reading comprehension abilities.

On p. 5, end of last paragraph before “Current Study” section, the last sentence is unclear. The authors describe an L1-underdevelopment explanation for absence of L1-L2 transfer; however, for the previously described studies where such transfer was occurring (e.g., the Chinese-English transfer study in the previous paragraph), children were much younger (and so could be expected to have even less developed L1). This seems like a contradiction that should be explained briefly, for example, to state that there are open questions on the impact of L1 and L2 proficiency on cross-language transfer in L2 reading comprehension.

At the end of the top paragraph on p. 6, it would be helpful to have some rationale about why it was important to focus on cross-language transfer at the point of the mediators (that is, that the association between SA in L1 and L2 reading comprehension is mediated by (an indirect effect through) L2 word reading and vocab and not L1 word reading and vocab).

In the Method, 

-       Indicate “second” instead of “s” for participant grade.

-       Was there a check on any possible influence of the third language on performance and outcomes of the six children with non-English home languages? How was this handled in the analysis – could these have influenced the patterns of results?

-       Indicate the units of analysis for each measure – raw scores or standard/scaled scores?

-       The use of an informal translation of Gates English reading comprehension test might be problematic – what was the reasoning behind using this and not an alternative French reading comprehension measure that was standardized (like the Compréhension de lecture subtest of the WIAT-II)? What was the unit of analysis for this and other measures?

-       Were there any missing data or incomplete data; if so how were these handled?

-       Why was phonological working memory from the CTOPP-2 (a measure of phonological short-term memory) used, and not a more-informative measure like Elision as a measure of phonological awareness? Also, replace “Complete” with “Comprehensive”.

-       The demographic information is given in the Participants section, and so the detailed description of the questionnaire can be shortened.

-       Be more specific about the number of sessions per participant in Procedures (give the range if more than one session for some).

Results

-       Don’t repeat reliability and other information in the tables that were reported in the text (or vice-versa).

-       Second paragraph: you might want to say something about the relatively low French SA scores, compared to English SA even at this point; fix the t-statistics (“t(67)”; “p < .001”). 

-       End of “Correlations” section – delete “moderately”.

-       Table 2 needs a bit of revision – some added explanation of the analytic technique used should be provided to ensure the reader does not interpret results as though they reflect a simple hierarchical regression rather than the stepwise – that is, indicate clearly the points at which IVs were eliminated. Also, NVR results are not reported here, and so the reference to this should be deleted in the notes (this seems to be erroneously identified as “MAT”).

-       Figures 2 and 3 – indicate standardized coefficients (if this is the case).

The discussion of the results – especially the differential patterns for direct and indirect influences – was really well done and provided excellent insights on what the results tell us about the relationship between SA and L2 reading comprehension. I’d like to see more on the discussion of the fourth question: Why was it the case that French vocabulary was not significant in the cross-language analysis? Was there stronger facilitation from English SA that made French vocabulary redundant, or not useful in French RC?

In the limitations, the authors should say something about the potential problems in using an informal translation of an existing English language reading comprehension measure, and also on the validity of the Gates reading comprehension measure, given its heavy reliance on decoding skills (especially for understanding the multiple-choice items). The suggestions to focus on age and developmental issues (especially related to language proficiency) for further research are apt. Despite the “moving target” involving L1 and L2 proficiency, this raises lots of interesting questions, e.g., what would happen in cases of “balanced bilingualism" (equal L1 and L2 proficiency) as a first language, and whether there are complementary or competing processes depending on target RC language. Same with structural similarities/differences pertinent to SA depending on language pairings. In the Conclusion, given that structural similarities, relative L1 and L2 proficiencies, and Immersion contexts were not manipulated and explicitly tested, the wording of the third last paragraph should be more carefully constructed (e.g., “consistent with” instead of “suggest”). In the last sentence “understanding” is ambiguous – maybe change to “children’s understanding of”; replace “enables” with “could enable”.

The references should be checked for completeness. One reference that features quite prominently in the discussion of syntactical bootstrapping (Babineau et al., 2019) is cited on several occasions but does not appear in the references. Is this De Carvalho, Alex, et al. “Studying the Real-Time Interpretation of Novel Noun and Verb Meanings in Young Children.” Frontiers in Psychology, vol. 10, 2019, https://doi.org/10.3389/fpsyg.2019.00274?

Just a thought for consideration: I wonder how L1 reading comprehension skill might have been influenced by L1 and L2 word reading and vocab abilities. Although not measured, some discussion about this would help to back up speculation and resolve some of the reported limitations especially regarding variability in L2 proficiency – perhaps setting the context in discussing the extent to which results have replicated previous literature on within- and cross-language influences on L2 reading comprehension would be helpful.

Author Response

(The authors gave the same response as above.)

Round 2

Reviewer 1 Report

I am satisfied that the authors sufficiently addressed my comments in the previous round and would be in favour of publication. 

Author Response

Reviewer 1

  1. I am satisfied that the authors sufficiently addressed my comments in the previous round and would be in favour of publication.

Response 1:

Thank you so much for your previous feedback! We really appreciate your guidance in improving this paper.

Reviewer 2 Report

Manuscript number – languages-1844676_R1

Title -  Syntactic Awareness and Reading Comprehension in Emergent Bilingual Children

The authors addressed my concerns quite nicely and have elaborated on the key issues expertly and clearly, and so the current version is ready for publication pending some minor concerns that would fall more under the category of copy-editing issues rather than revisions. These do need to be addressed for a final polished version:

The theoretical and educational implication section was a good addition, which places the findings of direct and indirect influences onto a clear theoretical foundation. A definition and example of sociolinguistic experience that influences cross-language transfer should be provided (e.g., informal communicative interactions among peers), as this term can be interpreted in various ways. This would eliminate potential ambiguity.

While I agree that the focus on SA would be muddied by a detailed discussion of MA, readers might wonder why MA was not included as a measure to establish SA as a unique contributor to RC. It might be helpful to mention that SA has been found to be a unique predictor of reading comprehension in previous studies that include morphological awareness, or that analysis of RC with word reading as a predictor (as was done in the current study) results in a nonsignificant impact of MA while retaining a significant role of SA, so word reading acts as a proxy for MA. This would reassure readers that including MA in the analysis would not likely have changed the pattern of findings. At least two cited studies report such findings (e.g., Deacon & Keiffer, 2018, and Tong et al., 2021).

The authors indicate in the Method section that the translation of the RC measure was created for the current study. However, in their responses to reviewer comments they indicate that the translation was used in previous studies. If this is the case, then this should be noted in the manuscript, and at least one of the citations given in the authors' responses should be included.

The revised version indicates that example items of the SA measures appear in an appendix, but this was not included in the manuscript -- this needs to be added.

In Table 1, confirm the SE values for Skewness and Kurtosis -- they are identical all the way through, which seems unlikely.

Double-check the coefficient values in the Figures for the links between English/French SA and French reading and vocab -- these values seem extremely large especially if they are standardized and given the relative values of the correlations.

Check for missing references:

e.g., Chung et al., 2019; Leider et al., 2013; Proctor et al., 2012; Swanson et al., 2008

Author Response

Reviewer 2

  1. The authors addressed my concerns quite nicely and have elaborated on the key issues expertly and clearly, and so the current version is ready for publication pending some minor concerns that would fall more under the category of copy-editing issues rather than revisions. These do need to be addressed for a final polished version:

Response 2: Thank you so much for your final suggestions! They were all helpful and will be addressed in detail below.

  1. The theoretical and educational implication section was a good addition, which places the findings of direct and indirect influences onto a clear theoretical foundation. A definition and example of sociolinguistic experience that influences cross-language transfer should be provided (e.g., informal communicative interactions among peers), as this term can be interpreted in various ways. This would eliminate potential ambiguity.

Response 3: This is an excellent point. You’re correct that we only gave this definition in the literature review and did not include it in the discussion. We have now added a few examples in the discussion section, please see lines 612-614.

  1. While I agree that the focus on SA would be muddied by a detailed discussion of MA, readers might wonder why MA was not included as a measure to establish SA as a unique contributor to RC. It might be helpful to mention that SA has been found to be a unique predictor of reading comprehension in previous studies that include morphological awareness, or that analysis of RC with word reading as a predictor (as was done in the current study) results in a nonsignificant impact of MA while retaining a significant role of SA, so word reading acts as a proxy for MA. This would reassure readers that including MA in the analysis would not likely have changed the pattern of findings. At least two cited studies report such findings (e.g., Deacon & Kieffer, 2018, and Tong et al., 2021).

 Response 4: Thank you for this point! We have now added this point to the limitation section. Specifically, we point out that it is a limitation not to include other aspects of metalinguistic awareness, such as morphological awareness, in our study. Following the reviewer's suggestion, we present the two previous studies (e.g., Deacon & Kieffer, 2018, and Tong et al., 2021) to indicate that the results might stay the same with this additional control variable, please see lines 637-640.

  1. The authors indicate in the Method section that the translation of the RC measure was created for the current study. However, in their responses to reviewer comments they indicate that the translation was used in previous studies. If this is the case, then this should be noted in the manuscript, and at least one of the citations given in the authors' responses should be included.

 Response 5: Thank you for pointing out this discrepancy, you’re right that we should have been more clear in the manuscript. We have now changed it, please see lines 345-347.

  1. The revised version indicates that example items of the SA measures appear in an appendix, but this was not included in the manuscript -- this needs to be added.

 Response 6: Thank you for catching this, we apologize for the oversight. This has now been added, please see Appendix A.

  1. In Table 1, confirm the SE values for Skewness and Kurtosis -- they are identical all the way through, which seems unlikely.

 Response 7: Thank you for this point! Upon further investigation we realized that the standard errors for skewness and kurtosis are solely functions of the sample size, regardless of the values of the statistics themselves. So for a given group, descriptive statistics for all variables will be based on the same number of cases. We have accordingly removed the S.E. columns and indicated the values in the text, please see lines 408-410,

  1. Double-check the coefficient values in the Figures for the links between English/French SA and French reading and vocab -- these values seem extremely large especially if they are standardized and given the relative values of the correlations.

 Response 8: Thank you for this point! After further research, the previous coefficients produced by PROCESS were indeed unstandardized. They have now been standardized and inserted in Figures 1 and 2.

Check for missing references:

e.g., Chung et al., 2019; Leider et al., 2013; Proctor et al., 2012; Swanson et al., 2008

Response 9: Thank you so much for catching this! All of the references have now been updated, and we really appreciate your attention to detail.